| Open Peer Review | Virology | New-Data Letter

# Disulfiram inhibits poxvirus extracellular virus production by targeting the palmitoylation sites on F13

Ting Xu,[1,2] Junwen Luan,[2] Yao Hou,[2] Leiliang Zhang[1,2]

**KEYWORDS** palmitoylation, F13, poxvirus, drug, disulfiram

Mpox virus (MPXV) has attracted significant attention due to its increasing occurrence since 2022. Tecovirimat (ST-246) is a well-known inhibitor that targets the poxvirus F13 protein family, effectively reducing the production of enveloped extracellular virus (EEV) but not intracellular mature virus (IMV) (1). A previous study has suggested that tecovirimat interacts with the phospholipase D enzymatic activity site of F13 (2). Recent findings indicate that tecovirimat functions as a molecular glue, promoting the dimerization of F13 (3). However, it remains unclear whether alternative targeting strategies for F13 exist.

Poxvirus F13 is known to be palmitoylated at cysteine (Cys, C) residues 185 and 186 (4). Disulfiram is an FDA-approved drug used in the treatment of alcohol use disorder (AUD) (5). Previous studies have indicated that disulfiram targets the cysteine residues of HCV NS5A to inhibit HCV replication (6). Therefore, we investigated whether disulfiram could interact with cysteine residues in F13.

We utilized AlphaFold3 to simulate the protein structures of palmitoylated MPXV F13, incorporating the palmitic acids on C185, C186, and C13 (predicted). Next, we employed AlphaFold3 to dock the interaction between palmitoylated F13 and disulfiram. Surprisingly, our analysis revealed that disulfiram interacts with the palmitic acids in F13 (Fig. 1A). Docking and visualization were performed using the Molecular Operating Environment (MOE) 2019. Disulfiram was observed to interact with leucine (Leu) 178, Cys181, serine (Ser) 182, palmitoylated Cys185, and palmitoylated Cys186 (Fig. 1B) in MPXV F13. Subsequently, we used AlphaFold3 to simulate the protein structures of MPXV F13 without palmitoylation and conducted docking studies between non-palmitoylated F13 and disulfiram (Fig. 1C). In this case, disulfiram was found to associate with phenylalanine (Phe) 52, Cys53, Cys120, Leu118, Ser135, threonine (Thr) 137, glycine (Gly) 139, Ser140, tryptophan (Trp) 279, and asparagine (Asp) 312 (Fig. 1D). Furthermore, we utilized AlphaFold3 to simulate the protein structures of palmitoylated vaccinia virus (VACV) F13, including the palmitic acids on Cys185 and Cys186 (Fig. 1E). Interestingly, disulfiram associated with alanine (Ala) 184, palmitoylated Cys185, and proline (Pro) 188 in VACV F13 (Fig. 1F).

To investigate the interaction between MPXV F13 and disulfiram, we conducted a cellular thermal shift assay (CETSA). Our results demonstrate that disulfiram increased the stability of F13, whereas it did not affect the stability of the F13 C185/C186S mutant (Fig. 1G and H). These findings suggest a specific association between disulfiram and F13 at the C185/C186 sites.

Next, we examined whether disulfiram inhibits VACV, which serves as a model system for studying poxviruses. Disulfiram significantly decreased the plaque size of the VACV Western Reserve (WR) strain in BSC-1 cells, with a calculated half maximal inhibitory concentration ($IC_{50}$) of 825.1 nmol/L (nM) (Fig. 2A). Cell Counting Kit-8 (CCK-8) assays demonstrated that the antiviral effects of disulfiram are not linked to cytotoxicity.

**Peer Reviewers** Elias A. Rahal, American University of Beirut, Riad El-Solh, Beirut, Lebanon; Ravindra P. Veeranna, Xavier University School of Medicine, Aruba, Aruba

Address correspondence to Leiliang Zhang, armzhang@hotmail.com.

Ting Xu, Junwen Luan, and Yao Hou contributed equally to this article. Author order was determined based on the time at which each author joined the project.

L.Z., T.X., and J.L. have a pending patent application for this study.

See the funding table on p. 4.

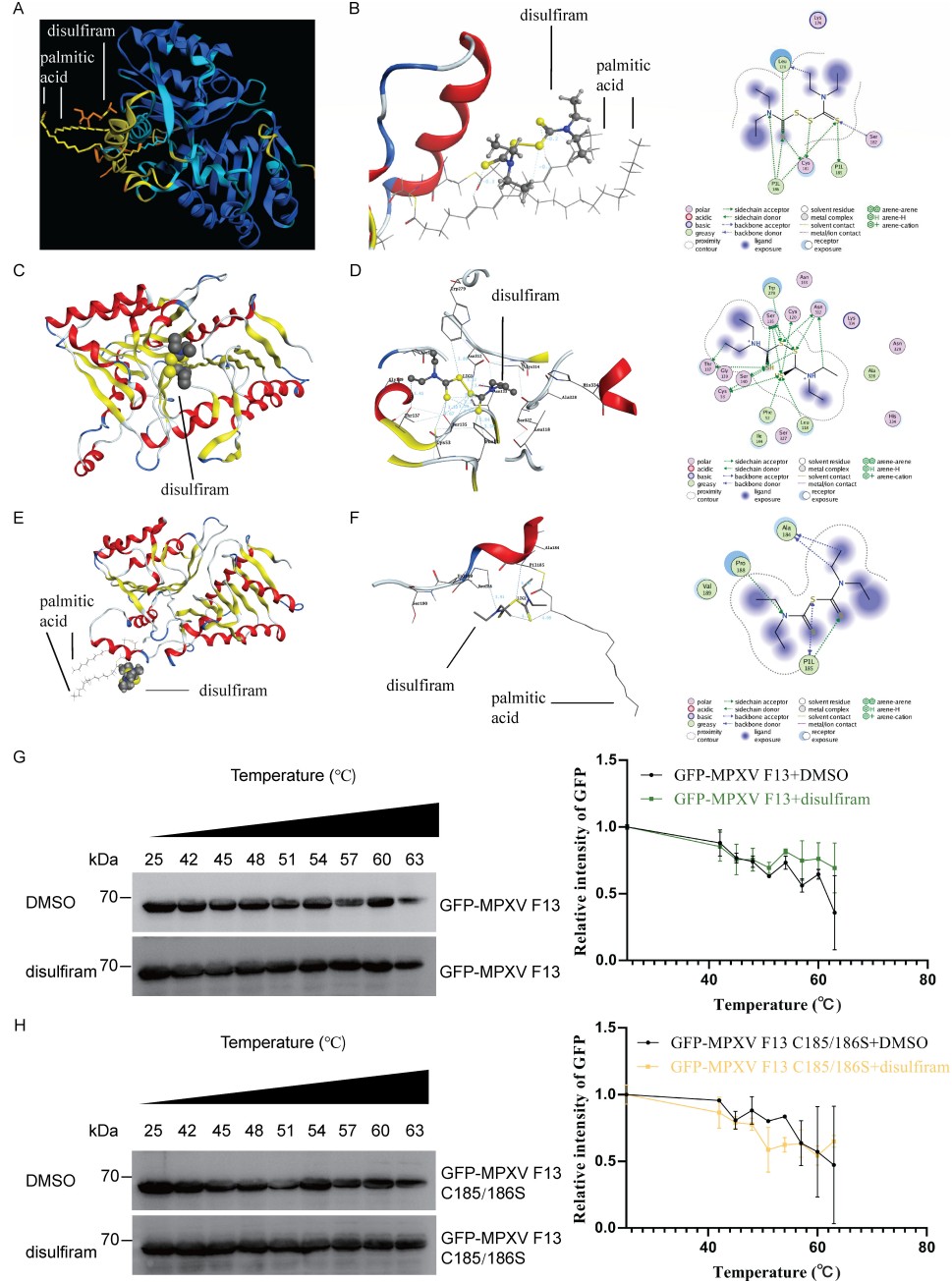

**FIG 1** Interaction between disulfiram and palmitoylated MPXV F13. (A) The association between disulfiram and the palmitic acids at C185 and C186 of palmitoylated MPXV F13 was identified by AlphaFold3. The structures of F13 from MPXV_USA_2022_MA001 virus strain and its palmitoylated form at C13, C185, and C186 were predicted using AlphaFold3. (B) Analysis of the specific binding sites of disulfiram to palmitoylated MPXV F13 using MOE2019. (C) The association between disulfiram and non-palmitoylated MPXV F13 was identified by AlphaFold3. (D) Analysis of the specific binding sites of disulfiram to non-palmitoylated MPXV F13 using MOE2019. (E) The association between disulfiram and the palmitic acids at C185 of palmitoylated VACV F13 was identified by AlphaFold3. The structures of F13 from the VACV WR strain and its palmitoylated form at C185 and C186 were predicted using AlphaFold3. (F) Analysis of the specific binding sites of disulfiram to palmitoylated VACV F13 using MOE2019. (G) 293T cells were transfected with GFP-MPXV F13 for 24 hours, after which the cells were harvested. Following three freeze-thaw cycles, the supernatant was collected by centrifugation, and 10 µM disulfiram or DMSO was added to the supernatant. The mixture was then heated to the specified temperatures before adding the loading buffer for western blot analysis ($n = 3$). Relative protein levels of GFP-MPXV F13 were quantified using ImageJ. (H) 293T cells were transfected with GFP-MPXV F13 C185/C186S for 24 hours, and the cells were subsequently harvested. After

**Fig 1 (Continued)**

undergoing three freeze-thaw cycles, the supernatant was obtained through centrifugation, to which either 10 µM disulfiram or DMSO was added. The mixture was then heated to the designated temperatures prior to the addition of loading buffer for western blot analysis ($n = 3$). The relative protein levels of GFP-MPXV F13 were quantified using ImageJ.

To further investigate the VACV life cycle, we concentrated on assessing the titers of IMV and EEV following disulfiram treatment. Notably, disulfiram inhibited EEV production ($IC_{50} = 7,125$ nmol/L) without impacting IMV production in Huh7.5.1 cells (Fig. 2B). A similar inhibition of EEV production was observed in HeLa cells ($IC_{50} = 3,022$ nmol/L) (Fig. 2C).

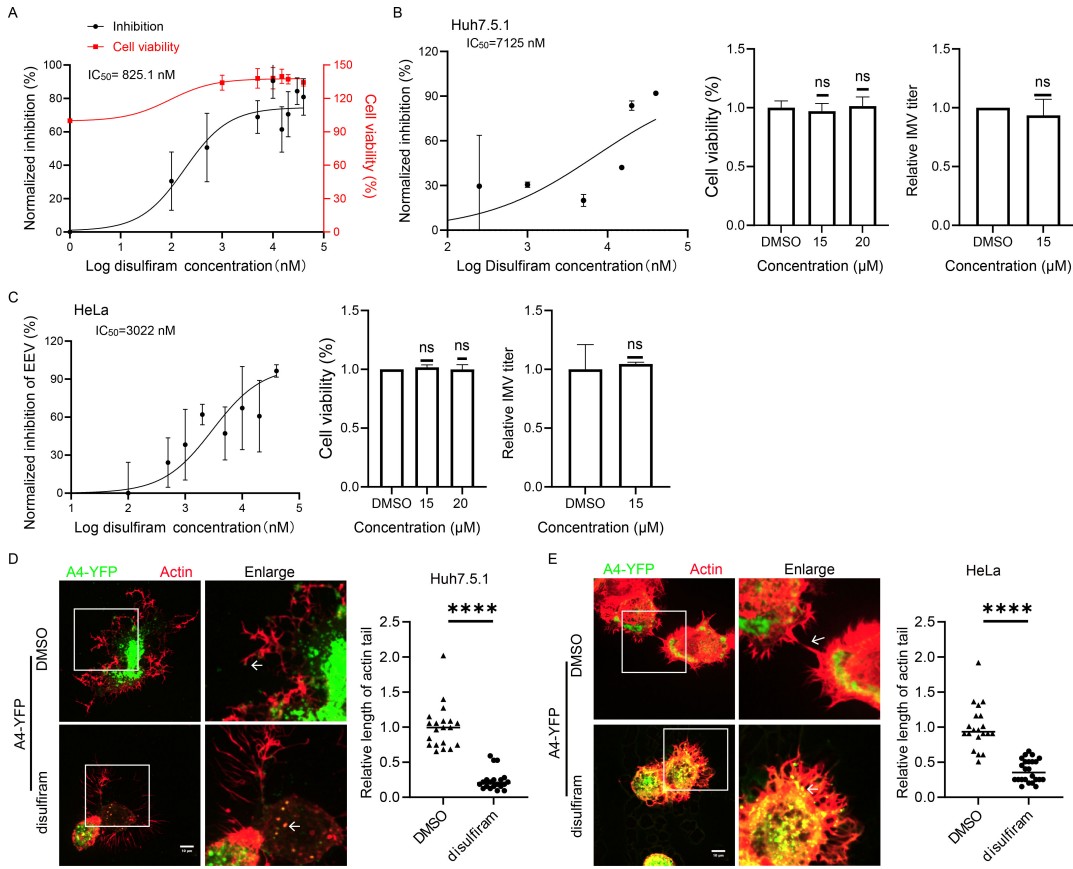

**FIG 2** Disulfiram inhibits the production of VACV EEV. (A) BSC-1 cells were infected with VACV WR, and after 2 h, fresh media containing different concentrations of disulfiram were added. Cells were collected 52 h post-infection and stained with 0.1% crystal violet. For the cell viability assay, BSC-1 cells were seeded in 96-well plates and grown to 50% confluence before treatment with various concentrations of disulfiram. After 52 h, CCK-8 reagent was added and incubated for 2 h. Absorbance was measured using a microplate reader, and $IC_{50}$ and $CC_{50}$ values were calculated using GraphPad Prism ($n = 3$). (B) Huh7.5.1 cells were infected with VACV WR (MOI = 3), and after 2 h, fresh media containing different concentrations of disulfiram were added for another 22 h. Supernatants and cells were collected to determine the viral titer and calculate the $IC_{50}$ value. Huh7.5.1 cells were seeded in 96-well plates and grown to 50% confluence before treatment with various concentrations of disulfiram. After 52 hours, CCK-8 reagent was added and incubated for 2 h. Absorbance was measured using a microplate reader ($n = 3$). Statistical analysis was conducted with GraphPad Prism ($t$-test). ns, $P > 0.05$. (C) HeLa cells were infected with VACV WR (MOI = 3), and after 2 h, fresh media containing different concentrations of disulfiram were added for another 22 h. Supernatants and cells were collected to determine the viral titer and calculate the $IC_{50}$ value. HeLa cells were seeded in 96-well plates and grown to 50% confluence before treatment with various concentrations of disulfiram. After 52 h, CCK-8 reagent was added and incubated for 2 h. Absorbance was measured using a microplate reader ($n = 3$). Statistical analysis was conducted with GraphPad Prism ($t$-test). ns, $P > 0.05$. (D and E) Huh7.5.1 (D) or HeLa (E) cells were infected with VACV A4-YFP at an MOI of 3. After 2 h, the medium was replaced with fresh media containing 15 µmol/L disulfiram. Cells were collected 22 h later, fixed with 4% PFA, and stained for actin with a working concentration of 80 nmol/L of phalloidin. Arrows denote actin tails. Scale bar: 10 µm. Quantification was performed using ImageJ ($n = 20$), and statistical analysis was conducted with GraphPad Prism ($t$-test). ****, $P < 0.0001$.

Considering the close relationship between EEV production and the dynamics of actin tails, which are crucial for viral egress, we evaluated the effects of disulfiram on actin tail lengths. Treatment with 15 µmol/L disulfiram led to a notable decrease in the lengths of actin tails in both Huh7.5.1 and HeLa cells (Fig. 2D and E). Although disulfiram may influence cellular pathways, our experiments show that the observed reduction in actin tail length correlates specifically with viral processes rather than broadly affecting host cell actin regulation. This specificity is further supported by our findings indicating that disulfiram's inhibitory effects on EEV formation do not result in widespread alterations to the overall structure of host cell actin.

Disulfiram is known to interact with thiol groups non-specifically. Although our findings indicate a specific association between disulfiram and F13, it is crucial to acknowledge that the observed effects may also stem from disulfiram's broader reactivity with thiol-containing molecules. Further studies are needed to elucidate the specific mechanisms underlying the observed inhibition and confirm that these effects are indeed mediated through the intended target, F13.

The F13 proteins of MPXV and VACV share 99% sequence identity (with only 3 amino acid differences among 372 residues), highlighting a high degree of structural conservation between these viral proteins. This significant conservation suggests that the mechanisms of action for disulfiram may be similar for both viruses. Given that disulfiram effectively interacts with VACV F13, it is likely that comparable interactions occur with MPXV F13. Our findings strongly justify the use of VACV in our functional assays while still addressing our hypothesis regarding MPXV F13.

In summary, our findings demonstrate that the FDA-approved drug disulfiram can interact with palmitic acids in poxvirus F13, inhibiting EEV formation. By elucidating the mechanisms underlying this inhibition, we can explore disulfiram's potential repurposing as an antiviral agent. Additionally, the insights gained from our study may inform the design of novel compounds that target the palmitoylation site on F13 or similar proteins in other viruses, ultimately advancing antiviral therapies. Our results indicate that targeting palmitic acids in poxvirus F13 represents a novel strategy against poxvirus, including MPXV. Further structural studies and clinical investigations are needed to translate our findings into actionable strategies for managing poxvirus outbreaks.

## ACKNOWLEDGMENTS

This work was supported by grants from National Natural Science Foundation of China (82272306), Taishan Scholars Program (tstp20221142), and Joint Innovation Team for Clinical & Basic Research (202409).

Ting Xu: Investigation; Junwen Luan: Software; Yao Hou: Investigation; Leiliang Zhang: Conceptualization, Writing – original draft.

## AUTHOR AFFILIATIONS

[1]Department of Clinical Laboratory Medicine, The First Affiliated Hospital of Shandong First Medical University & Shandong Provincial Qianfoshan Hospital, Jinan, Shandong, China

[2]Department of Pathogen Biology, School of Clinical and Basic Medical Sciences, Shandong First Medical University & Shandong Academy of Medical Sciences, Jinan, Shandong, China

## AUTHOR ORCIDs

Leiliang Zhang http://orcid.org/0000-0002-7015-9661

## FUNDING

| Funder | Grant(s) | Author(s) |
| --- | --- | --- |
| Joint Innovation Team for Clinical & Basic Research | 202409 | Leiliang Zhang |

| Funder | Grant(s) | Author(s) |
|---|---|---|
| National Natural Science Foundation of China | 82272306 | Leiliang Zhang |
| Taishan Scholars Program | tstp20221142 | Leiliang Zhang |

## AUTHOR CONTRIBUTIONS

Ting Xu, Investigation | Junwen Luan, Software | Yao Hou, Investigation | Leiliang Zhang, Conceptualization, Writing – original draft

## ADDITIONAL FILES

The following material is available online.

### Open Peer Review

**PEER REVIEW HISTORY (review-history.pdf).** An accounting of the reviewer comments and feedback.

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
