## [Reviewer comments · Microbiology Spectrum]

Microbiology Spectrum

Disulfiram inhibits poxvirus extracellular virus production by targeting the palmitoylation sites on F13

Ting Xu, Junwen Luan, Yao Hou, and Leiliang Zhang

Corresponding Author(s): Leiliang Zhang, Shandong First Medical University

Review Timeline:

Submission Date:	June 5, 2025
Editorial Decision:	June 11, 2025
Revision Received:	June 13, 2025
Editorial Decision:	July 10, 2025
Revision Received:	August 29, 2025
Accepted:	September 18, 2025

Editor: Biao He

Reviewer(s): Disclosure of reviewer identity is with reference to reviewer comments included in decision letter(s). The following individuals involved in review of your submission have agreed to reveal their identity: Elias A. Rahal (Reviewer #1); Ravindra P Veeranna (Reviewer #2)

Transaction Report:

DOI: <https://doi.org/10.1128/spectrum.01752-25>

Re: Spectrum01752-25 (Disulfiram: An FDA-approved drug as a novel anti-poxvirus strategy targeting the palmitoylation sites of F13)

Dear Dr. Leiliang Zhang:

Thank you for the privilege of reviewing your work. To facilitate the reviewer assignment process, I kindly request that you provide a brief abstract of the article within the submission system at your earliest convenience. This will enable us to efficiently identify suitable reviewers and proceed with the evaluation.

The submission has been returned to you for this update. Should you require any assistance in completing this step, please do not hesitate to contact me directly.

Revision Guidelines

Sincerely,
Biao He
Editor
Microbiology Spectrum

You have submitted your manuscript as a New-Data Letter. New Data Letters report new, concise findings that are not appropriate for Research Articles. New-Data Letters should include no more than 500 words (exclusive of references), and they should not include an Abstract. Please format your manuscript as a New Data Letter and resubmit. Kindly provide abstract in submission system and R2R alone and not in the manuscript.

--A: Thanks for your suggestion. The manuscript has been switched the previous version of New-Data Letters. We provided the abstract in submission system.

Re: Spectrum01752-25R1 (Disulfiram: An FDA-approved drug as a novel anti-poxvirus strategy targeting the palmitoylation sites of F13)

Dear Dr. Leiliang Zhang:

Thank you for the privilege of reviewing your work. Below you will find my comments, instructions from the Spectrum editorial office, and the reviewer comments.

Both reviewers expressed concerns that the current dataset is limited in scope, and the conclusions would be significantly strengthened by incorporating additional experiments and a more robust supporting discussion.

Revision Guidelines

Sincerely,
Biao He
Editor
Microbiology Spectrum

Reviewer #1 (Comments for the Author):

The New-Data Letter by Xu et al. explores the potential of disulfiram, an FDA-approved drug for alcohol use disorder, as a novel antiviral against poxviruses by targeting the palmitoylation sites (C185 and C186) on the F13 protein. The authors used AlphaFold3 and Molecular Operating Environment (MOE) to model interactions and performed in vitro assays with vaccinia virus

(VACV). I have the following concerns and suggestions:

- AlphaFold is not optimized for docking or small molecule interaction predictions; hence, including this undermines the study. MOE is more reliable in this respect.
- Experimental validation of the modeling is not performed, which is a major drawback. The authors should experimentally validate interactions via pull-down assays incorporating mutagenesis of C185/C186.
- Disulfiram is known to interact with thiol groups non-specifically. Therefore, the observed inhibition might be off-target.
- While the modeling data is with MPXV, the biological data is with VACV, which is another factor undermining the study. All functional data are from VACV. MPXV relevance is therefore inferred but not shown.
- 15 $\mu\text{mol/L}$ disulfiram was used for actin tail disruption which is much higher than the IC_{50} . Potential off-target cytoskeletal effects are possible. What is the justification for using this dose? Was a dose-response curve conducted?

Reviewer #2 (Comments for the Author):

Summary:

This brief communication explores the potential of disulfiram-an FDA-approved drug for alcohol use disorder-as an antiviral agent against poxviruses by targeting the palmitoylated cysteine residues of the F13 protein. The study combines in silico protein modeling (AlphaFold3 and MOE) and in vitro assays using VACV (a poxvirus model) to show that disulfiram interacts with palmitic acid-modified residues C185 and C186 on F13 and inhibits extracellular enveloped virus (EEV) production without affecting intracellular mature virus (IMV). The findings suggest disulfiram may act through a novel mechanism distinct from tecovirimat.

Major Comments:

1. The interaction between disulfiram and the F13 palmitoylation sites is inferred from docking studies. While suitable for this format, the authors should verify using AlphaFold3 to model the monkeypox virus F13 protein without palmitoylated residues considering the results that disulfiram interacts with Leucine 178 and serine 182 of F13.
2. The claim that disulfiram works via a mechanism different from tecovirimat is compelling. Authors should conduct comparative experiment of phenotypic effects (e.g., EEV suppression, IMV unaffected) or IC_{50} values.
3. Although the hypothesis targets MPXV F13, all in vitro testing was performed using VACV. Please clarify the structural conservation between VACV and MPXV F13 proteins and discuss any known cross-applicability of disulfiram between these viruses.
4. Disulfiram is broadly thiol-reactive. The manuscript should include a sentence or two discussing whether other viral or host targets may be affected, and how specificity toward F13 was inferred.
5. The reduction in actin tail length is noted as a phenotype. Could this be due to a more general cytoskeletal disruption? Consider discussing whether the effect is specific to viral egress or a broader effect of disulfiram on host cell actin regulation.

Minor Comments:

1. Consider refining the title for clarity and specificity:
"Disulfiram inhibits poxvirus extracellular virus production by targeting palmitoylation sites on F13"
2. For Figure 2, indicate the number of replicates and type of statistical test used (t-test, ANOVA?). Include full p-values if possible.
3. Use consistent terminology when referring to "palmitoylated residues" versus "palmitic acids".
4. The authors disclose a patent application, which is appropriate. Consider also briefly mentioning how this may relate to future translational research or drug development.

Reviewer #1 (Comments for the Author):

The New-Data Letter by Xu et al. explores the potential of disulfiram, an FDA-approved drug for alcohol use disorder, as a novel antiviral against poxviruses by targeting the palmitoylation sites (C185 and C186) on the F13 protein. The authors used AlphaFold3 and Molecular Operating Environment (MOE) to model interactions and performed in vitro assays with vaccinia virus (VACV). I have the following concerns and suggestions:

-AlphaFold is not optimized for docking or small molecule interaction predictions; hence, including this undermines the study. MOE is more reliable in this respect.

--A: Thank you for your valuable feedback. We would like to clarify that we utilized AlphaFold3 in our study, which has been specifically optimized for predicting interactions between proteins and small molecules, unlike AlphaFold2. This enhancement allows for more reliable predictions in the context of docking studies. We appreciate your insights and believe that our approach aligns with the aim of the research.

-Experimental validation of the modeling is not performed, which is a major drawback. The authors should experimentally validate interactions via pull-down assays incorporating mutagenesis of C185/C186.

--A: Thank you for your insightful comments. In our study, we utilized the Cellular Thermal Shift Assay (CESTA) to evaluate the interaction between F13 and disulfiram. Our results demonstrate that disulfiram significantly increased the stability of F13, while it did not affect the stability of the F13 C185/C186S mutant (Fig. 1G and 1H). These findings suggest a specific association between disulfiram and F13 at the C185/C186.

-Disulfiram is known to interact with thiol groups non-specifically. Therefore, the observed inhibition might be off-target.

--A: Thank you for your comment. Our CESTA demonstrated that disulfiram can stabilize F13, indicating a specific interaction between disulfiram and F13 (Fig. 1G). We acknowledge the potential for off-target effects, given that disulfiram is known to interact non-specifically with thiol groups. We have included a discussion regarding the off-target activity of disulfiram in Line 76-81.

“Disulfiram is known to interact with thiol groups non-specifically. While our findings indicate a specific association between disulfiram and F13, it is crucial to acknowledge that the observed effects may also stem from disulfiram's broader reactivity with thiol-containing molecules. Further studies are needed to elucidate the specific mechanisms underlying the observed inhibition and confirm that these effects are indeed mediated through the intended target, F13.”

-While the modeling data is with MPXV, the biological data is with VACV, which is another factor undermining the study. All functional data are from VACV. MPXV relevance is therefore inferred but not shown.

--A: Thank you for your thoughtful feedback. In our study, we not only performed docking analyses to investigate the interaction between disulfiram and MPXV F13, but we also examined the interaction between disulfiram and VACV F13, as presented in Fig. 1E and 1F. Our results indicate that disulfiram associates with the palmitoylated C185 residue of VACV F13. Given this interaction and to provide a robust functional analysis, we chose to use VACV for our functional assays. This approach allowed us to draw meaningful connections between the biological and modeling data.

- 15 $\mu\text{mol/L}$ disulfiram was used for actin tail disruption which is much higher than the IC_{50} . Potential off-target cytoskeletal effects are possible. What is the justification for using this dose? Was a dose-response curve conducted?

--A: Thank you for your comment. The actin tail is essential for the formation of extracellular enveloped virus (EEV), and we aimed for a more substantial disruption of this process. While using the IC_{50} dosage would yield a moderate reduction in EEV formation, we determined that a higher concentration of 15 $\mu\text{mol/L}$ disulfiram was necessary to achieve a more significant and consistent effect on actin tail disruption. This concentration was chosen based on preliminary experiments that suggested this dose effectively compromised actin tail dynamics, thus allowing us to investigate the impact on EEV formation more thoroughly. However, we recognize your concern regarding potential off-target cytoskeletal effects. Unfortunately, a

dose-response curve was not conducted in this study, but we appreciate your suggestion for future experiments to establish a clearer relationship between disulfiram concentration and its effects on actin dynamics.

Reviewer #2 (Comments for the Author):

Summary:

This brief communication explores the potential of disulfiram-an FDA-approved drug for alcohol use disorder-as an antiviral agent against poxviruses by targeting the palmitoylated cysteine residues of the F13 protein. The study combines in silico protein modeling (AlphaFold3 and MOE) and in vitro assays using VACV (a poxvirus model) to show that disulfiram interacts with palmitic acid-modified residues C185 and C186 on F13 and inhibits extracellular enveloped virus (EEV) production without affecting intracellular mature virus (IMV). The findings suggest disulfiram may act through a novel mechanism distinct from tecovirimat.

Major Comments:

1. The interaction between disulfiram and the F13 palmitoylation sites is inferred from docking studies. While suitable for this format, the authors should verify using AlphaFold3 to model the monkeypox virus F13 protein without palmitoylated residues considering the results that disulfiram interacts with Leucine 178 and serine 182 of F13.

--A: Thank you for your comments. We have utilized AlphaFold3 to model the MPXV F13 protein without palmitoylated residues, as illustrated in Fig. 1C. Our analysis showed that the non-palmitoylated F13 did not interact with Leucine 178 or Serine 182 (Fig. 1D). This finding provides further support for our initial docking studies and indicates that the palmitoylation of F13 plays a crucial role in facilitating disulfiram's interaction with these specific residues. We appreciate your suggestion to verify our results with AlphaFold3, and we believe this additional modeling strengthens the validity of our conclusions.

2. The claim that disulfiram works via a mechanism different from tecovirimat is compelling. Authors should conduct comparative experiment of phenotypic effects (e.g., EEV suppression, IMV unaffected) or IC₅₀ values.

--A: Thank you for your suggestion. As we previously demonstrated (<https://www.sciencedirect.com/science/article/pii/S2590053625000011>), the IC₅₀ value for tecovirimat in HeLa cells is 54.30 nM, which is significantly lower than that of disulfiram, which has an IC₅₀ of 3022 nM in the same cell line. This difference in potency supports our claim that disulfiram operates via a distinct mechanism compared to tecovirimat.

3. Although the hypothesis targets MPXV F13, all in vitro testing was performed using VACV. Please clarify the structural conservation between VACV and MPXV F13 proteins and discuss any known cross-applicability of disulfiram between these viruses.

--A: Thank you for your suggestion. In our study, we conducted docking analyses to investigate the interaction between disulfiram and MPXV F13, as well as examining its interaction with VACV F13, as illustrated in Fig. 1E and 1F. Our results indicate that disulfiram specifically associates with the palmitoylated C185 residue of VACV F13. The F13 proteins of MPXV and VACV share 99% sequence identity (with only 3 amino acid differences among 372 residues), highlighting a high degree of structural conservation between these viral proteins. This significant conservation suggests that the mechanisms of action for disulfiram may be similar for both viruses. Given that disulfiram effectively interacts with VACV F13, it is likely that comparable interactions occur with MPXV F13. Our findings strongly justify the use of VACV in our functional assays while still addressing our hypothesis regarding MPXV F13.

We have expanded our discussion on structural similarity and its functional implications in Line 82-88 of the manuscript. We appreciate your insights, as they have helped us clarify the relevance of our study in the context of both MPXV and VACV.

4. Disulfiram is broadly thiol-reactive. The manuscript should include a sentence or two discussing whether other viral or host targets may be affected, and how specificity toward F13 was inferred.

--A: Thanks for your comment. We have added some words on other potential targets in Line 76-81.

“Disulfiram is known to interact with thiol groups non-specifically. While our findings indicate a specific association between disulfiram and F13, it is crucial to acknowledge that the observed effects may also stem from disulfiram's broader reactivity with thiol-containing molecules. Further studies are needed to elucidate the specific mechanisms underlying the observed inhibition and confirm that these effects are indeed mediated through the intended target, F13.”

5. The reduction in actin tail length is noted as a phenotype. Could this be due to a more general cytoskeletal disruption? Consider discussing whether the effect is

specific to viral egress or a broader effect of disulfiram on host cell actin regulation.

--A: Thank you for your comment. The actin tail is essential for the formation of extracellular enveloped virus (EEV), and a reduction in actin tail length directly inhibits EEV formation. Our data indicate that this effect is specific to viral egress rather than a more general disruption of host cell actin dynamics by disulfiram. We have included additional discussion on this topic in Line 70-75 of the manuscript to address these points more comprehensively.

“While disulfiram may influence cellular pathways, our experiments show that the observed reduction in actin tail length correlates specifically with viral processes rather than broadly affecting host cell actin regulation. This specificity is further supported by our findings indicating that disulfiram's inhibitory effects on EEV formation do not result in widespread alterations to the overall structure of host cell actin.”

Minor Comments:

1. Consider refining the title for clarity and specificity:

"Disulfiram inhibits poxvirus extracellular virus production by targeting palmitoylation sites on F13"

--A: Thanks for your comment. We have refined the title as “Disulfiram inhibits poxvirus extracellular virus production by targeting the palmitoylation sites on F13”.

2. For Figure 2, indicate the number of replicates and type of statistical test used (t-test, ANOVA?). Include full p-values if possible.

--A: Thanks for your comment. In revised manuscript, we have included the number of replicates and type of statistical test used (t-test)

3. Use consistent terminology when referring to "palmitoylated residues" versus "palmitic acids".

--A: Thanks for your comment. We have corrected the terminology to "palmitic acids".

4. The authors disclose a patent application, which is appropriate. Consider also briefly mentioning how this may relate to future translational research or drug development.

--A: Thank you for your comment. We have added a brief discussion on how our patent application relates to future translational research and drug development in Line 90-94.

“By elucidating the mechanisms underlying this inhibition, we can explore disulfiram’s potential repurposing as an antiviral agent. Additionally, the insights gained from our study may inform the design of novel compounds that target the

palmitoylation site on F13 or similar proteins in other viruses, ultimately advancing antiviral therapies.”

We appreciate your suggestion to address this important aspect, as it underscores the potential clinical relevance of our work.

Re: Spectrum01752-25R2 (**Disulfiram inhibits poxvirus extracellular virus production by targeting the palmitoylation sites on F13**)

Dear Dr. Leiliang Zhang:

I am glad to inform you that your manuscript has been accepted, and I am forwarding it to the ASM production staff for publication. Your paper will first be checked to make sure all elements meet the technical requirements. ASM staff will contact you if anything needs to be revised before copyediting and production can begin. Otherwise, you will be notified when your proofs are ready to be viewed.

Sincerely,
Biao He
Editor
Microbiology Spectrum

Reviewer #1 (Comments for the Author):

The authors have relatively adequately responded to my previous comments and applied changes where they were capable.